# Force Plate-Derived Countermovement Jump Normative Data and Benchmarks for Professional Rugby League Players

**DOI:** 10.3390/s22228669

**Published:** 2022-11-10

**Authors:** John J. McMahon, Nicholas J. Ripley, Paul Comfort

**Affiliations:** 1Centre for Human Movement and Rehabilitation, University of Salford, Salford M6 6PU, UK; 2School of Medical and Health Sciences, Edith Cowan University, Joondalup, WA 6027, Australia

**Keywords:** vertical jump, force platform, Z-score, reference data, benchmarking, data visualization

## Abstract

The countermovement jump (CMJ) is an important test in rugby league (RL), and the force plate is the recommended assessment device, as it permits the calculation of several variables that explain jump strategy, alongside jump height. The purpose of this study was to produce normative CMJ data and objective benchmarks for professional RL forwards and backs. Normative data for jump height, modified reactive strength index, and jump momentum are provided for 121 professional RL players (66 forwards and 55 backs) who completed CMJ testing on a portable force plate during preseason training. Standardized T-scores (scaled from 0 to 100) were calculated from the respective positional group mean and standard deviation to create CMJ performance bands that were combined with a qualitative description (ranging from extremely poor to excellent) and a traffic light system to facilitate data interpretation and objective benchmark setting by RL practitioners. The jump height and modified reactive strength index benchmarks were larger for the lighter backs, whereas the jump momentum benchmarks were larger for the heavier forwards. The presented novel approach to compiling and presenting normative data and objective benchmarks may also be applied to other data (i.e., from other tests or devices) and populations.

## 1. Introduction

The bilateral countermovement jump (CMJ) is the most common test of lower body neuromuscular function in peer-reviewed studies involving athletes from the football codes [1]. According to a systematic review of practitioner surveys, the third most common type of physical testing conducted by strength and conditioning coaches, behind body composition and strength, is muscular power [2]. In rugby union, the most common test of muscular power conducted by strength and conditioning coaches is the CMJ, with jump height being the most common metric [3]. The results of a more recent survey conducted with Argentinean strength and conditioning coaches working in rugby union showed that 35% used electronic jump mats to assess jump height, whereas 17% used force plates [4], with only force plates permitting the calculation of impulse applied to the body’s center of mass from the initiation of the jump through to take-off. In contrast, in professional soccer, half of strength and conditioning coaches used force plates to assess their athletes with the other half using electronic jump mats [5]. The rise in force plate assessments of the CMJ across the football codes is likely to continue as more commercially available force plate systems are validated against industry gold standard force plate systems [6,7]. Force plate assessments of the CMJ involving rugby league athletes have featured in many recent peer-reviewed experimental studies [8,9,10,11]. This is because the CMJ has been suggested to be an important test of lower body neuromuscular performance for rugby league players [12], in part owing to greater CMJ heights being correlated (*r* = 0.38–0.62) with better sprint and tackle performances [13,14], while increasing the likelihood of being selected to compete in professional matches [15] and distinguishing senior from academy players [16].

In rugby league, players can be broadly categorized as competing in either the forward or back positional groups. A recent study showed that CMJ height, modified reactive strength index, and both mean and peak propulsion power were significantly larger (small–moderate effect sizes) for professional rugby league backs compared with forwards [8]. In the study, scale of reference values (percentiles) were accordingly reported for each of these four CMJ metrics for both the forwards and backs independently [8]. A rationale for calculating jump take-off momentum (described simply as jump momentum herein) as part of routine CMJ testing of rugby league players was more recently presented [9]. More recently still, the authors of another study reported that only the CMJ outcome variables, specifically jump height (moderate effect size) and jump momentum (large effect size), differentiated between rugby league forwards and backs rather than modified reactive strength index and a selection of strategy variables [10]. It is worth noting that each of these studies involved data collection within the first two weeks of the preseason period, which may not reflect the rugby league players’ best CMJ performance but provides a comparable stage of the year when conducting multi-club studies. Furthermore, CMJ mean and peak propulsion power were not included in the latter study [10] and a possible reason for the nonsignificant positional differences in modified reactive strength index may have been the relatively small sample size (*n* = 27) tested for each positional group [10]. Nonetheless, from the collective results of these studies [8,10], it may be suggested that a CMJ scale of reference values for rugby league forwards and backs should be independently determined (i.e., position-specific) and based on primary outcome variables, such as jump height, modified reactive strength index, and jump momentum. 

Objective benchmarks (i.e., target setting) cannot be determined for any test metric without the existence or creation of cohort-specific normative data sets. Standardized scores, namely Z-scores, are calculated as multiples of the standard deviation in relation to the mean [17,18]. Specifically, 99.7% of individual data points reside within three standard deviations of the mean, assuming a normal data distribution, which results in a typical Z-score range from −3 to 3, with 0 representing the mean value. Z-scores are commonly calculated for a variety of force plate metrics by commercial force plate software providers. Z-scores allow practitioners to understand how well an individual athlete performs in relation to the squad mean (i.e., how many standard deviations above or below the squad mean each individual athlete’s score is) for any given force plate metric and, therefore, make informed training program decisions based on the identified physical strengths and deficits. T-scores, which are an alternative form of standardized score [19], may be calculated by multiplying Z-scores by 10 and adding 50 to provide a more intuitive value for athletes and sports coaches (scaled from 0 to 100 (or 20–80 when considering a typical three standard deviation limit about the mean), with a score of 50, rather than 0, equaling the mean). This notion is visually depicted in Figure 1, where corresponding Z-scores and T-scores for an assumed normally distributed (‘bell curve’) data set are shown. 

The purpose of this study was to compile normative data for CMJ outcome metrics attained by professional rugby league forwards and backs. A limitation of the earlier study in which CMJ performance standards of rugby league players was explored is that only percentiles were presented [8]. Additionally, in the previous study, the authors did not include any recommendations on how to qualitatively interpret and visually report CMJ standards (i.e., benchmarks) to rugby league athletes and coaches. Following on from case study examples provided in the most recently published study [10], the aim of the present study was to also provide examples of how to qualitatively interpret and report benchmarks for CMJ outcome metrics to rugby league athletes and coaches. 

## 2. Materials and Methods

A total of 121 rugby league players from the English Super League (*n* = 59) and Rugby League Championship (*n* = 62), comprising 66 forwards (25 ± 4 years, 1.84 ± 0.06 m, 102 ± 9 kg) and 55 backs (25 ± 4 years, 1.81 ± 0.06 m, 90 ± 9 kg), participated in this study by attending a single testing session during the first week of their respective preseason training periods. The results of a recent study conducted with professional rugby union players revealed no differences in the reliability statistics for a range of CMJ outcome and strategy metrics recorded across four separate testing sessions during the first week of the preseason period [20]; thus, a single testing session was deemed appropriate. Participants had previous experience of performing CMJs in line with the protocols discussed in the procedures section. Although a priori sample size estimations for the generation of cohort-specific normative data sets are difficult to determine, it was previously suggested that a minimum of 50 participants per population group should be sufficient in most cases, with greater than 75 participants providing no additional benefit [21]. Written informed consent was provided prior to testing, which was preapproved by the institutional review board and conformed to the World Medical Association’s Declaration of Helsinki.

Following a brief warm-up, inclusive of dynamic stretching and submaximal jumping (5 × 1 repetitions of single effort CMJs and 2 × 5 repetitions of repeated CMJs), participants performed three recorded maximal-effort CMJs to their preferred countermovement depth [22], interspersed by an ~1 min rest. They were instructed to “jump as fast and as high as possible”, while constantly keeping their hands on their hips. The participants were informed that the “jump as fast” part of the verbal cue referred to them performing the countermovement (i.e., downward) and propulsion (i.e., upward) phases of the CMJ as quickly as possible. Verbal cues were standardized due to their reported influence on CMJ force–time characteristics [23,24]. 

Ground reaction forces during the maximal effort CMJs were sampled at 1000 Hz for 5 s using a single portable force plate (type 9286AA, Kistler Instruments Inc., Amherst, NY, USA), which was interfaced with a laptop running BioWare software (version 5.11, Kistler Instruments Inc., Amherst, NY, USA) to acquire the data. The force plate was zeroed between CMJ trials. Participants stood upright and motionless for the first second of data collection [25,26] to enable calculation of body weight (in Newtons as the vertical force averaged over 1 s) and body mass (in kilograms as the body weight divided by gravitational acceleration). The vertical component of the ground reaction force–time data for each CMJ trial was exported in a raw form as a text file and subsequently analyzed using a customized Microsoft Excel spreadsheet (Microsoft Corp., Redmond, WA, USA). Previous research supports the analyses of raw over filtered CMJ force–time data [27]. 

For each individual CMJ trial, net force was calculated by subtracting body weight from every force sample in the force–time series (each 0.001 s for the 5000 recorded samples). Center-of-mass velocity was then determined by dividing net force by body mass on a sample-by-sample basis and then integrating the product using the trapezoid rule [26]. Center-of-mass displacement was determined by integrating the velocity data at each time point, also using the trapezoid rule [26]. The onset of movement was identified in line with current recommendations for CMJs [25]. The instant of take-off was identified when force fell below a threshold equal to five times the standard deviation of the flight phase force [7,28,29]. The standard deviation of the flight phase force was calculated across the middle 50% of the flight phase duration when the force plate was unloaded [7,28]. 

The three CMJ jump variables of interest were then calculated. Take-off velocity was calculated as the center-of-mass velocity at the instant of take-off. Jump height was derived from said velocity at take-off [26]. The modified reactive strength index was calculated as jump height divided by time to take-off (which was the time between onset of movement and take-off). Mean and peak propulsion power were not included in this study, despite them being reported in a previous similar study [8]. This is because mean propulsion power is highly correlated (*R*^2^ = 0.81) with the modified reactive strength index [30] and peak propulsion power is highly correlated (*R*^2^ = 0.69) with jump height [31]. Jump momentum was calculated by multiplying take-off velocity by body mass [9]. The mean of each variable across the three recorded trials was statistically analyzed. 

### Statistical Analyses

The CMJ data were separated for the forwards and backs. The CMJ metrics were used for the normative data compilation and were normally distributed based on the results of a Shapiro–Wilk test. To justify the separation of the forwards’ and backs’ data, an independent *t*-test was performed for each jump variable and height and body mass data (alpha level < 0.05) along with the Hedges’ *g* effect size calculation (and corresponding lower and upper 95% confidence interval (CI_95_)). Hedges’ *g* effect sizes were interpreted as trivial (≤0.19), small (0.20–0.49), moderate (0.50–0.79), or large (≥0.80) [32]. A series of two-way mixed-effects model (absolute agreement and average measures) intraclass correlation coefficients (ICCs), along with the upper and lower CI_95_, were used to determine the relative (i.e., rank-order) between-trial reliability of each variable. Based on the lower-bound CI_95_ of the ICC estimate, <0.5, between 0.5 and 0.75, between 0.75 and 0.90, and >0.90 were indicative of poor, moderate, good, and excellent relative reliability, respectively [33]. These statistical tests were conducted in SPSS (version 27; SPSS Inc., Chicago, IL, USA). 

To create benchmarks for the presented jump data, T-score performance bands were created and allocated qualitative descriptions (see the following text in brackets), ranging from extremely poor to excellent, as follows: <20 (extremely poor), ≥20–≤30 (very poor), >30–≤40 (poor), >40–≤45 (below average), >45–≤55 (average), >55–≤60 (above average), >60–≤70 (good), >70–≤80 (very good), and >80 (excellent). A traffic light system approach was applied to the T-score performance bands to compliment the allocated qualitative descriptions (see Figure 1) and thereby further ease data interpretation for the intended end user [34]. The compilation of normative data, construction of T-score performance bands and application of the traffic light system were performed using Microsoft Excel (Microsoft Corp., Redmond, WA, USA).

## 3. Results

Based on the lower bound CI_95_ of the ICC, jump height (ICC = 0.956 [CI_95_ range = 0.941–0.968]), modified reactive strength index (ICC = 0.953 [CI_95_ range = 0.937–0.966]) and jump momentum (ICC = 0.988 [CI_95_ range = 0.984–0.991]) showed excellent between-trial reliability. 

The position-specific data distributions for jump height, modified reactive strength index and jump momentum are shown in Figure 2, Figure 3 and Figure 4. 

The forwards were taller than the backs (1.84 ± 0.06 m vs. 1.81 ± 0.06 m, *p* = 0.005) with a moderate effect size (*g* = 0.55 [CI_95_ range = 0.15–0.95]). The forwards were also heavier than the backs (102 ± 9 kg vs. 90 ± 9 kg, *p* < 0.001) with a large effect size (*g* = 1.44 [CI_95_ range = 0.98–1.89]). The mean jump height was higher for the backs (36.4 ± 4.5 cm vs. 34.1 ± 4.2 cm, *p* = 0.005) with a moderate effect size (*g* = 0.50 [CI_95_ range = 0.13–0.87]). The mean modified reactive strength index was also higher for the backs (0.49 ± 0.08 vs. 0.44 ± 0.08, *p* = 0.003) with a moderate effect size (*g* = 0.55 [CI_95_ range = 0.18–0.93]). Contrastingly, the mean jump momentum was higher for the forwards (262.6 ± 25.5 kg·m/s vs. 240.3 ± 27.8 kg·m/s, *p* < 0.001) with a large effect size (*g* = 0.80 [CI_95_ range = 0.41–1.19]). 

The position-specific normative data and benchmarks (based on the T-score bands) for jump height, modified reactive strength index, and jump momentum are presented in Figure 5. These benchmarks ranged from extremely poor to excellent, in line with the previously described T-score performance bands. 

## 4. Discussion

The purpose of this study was to compile normative data and create benchmarks for three primary CMJ outcome metrics attained by professional rugby league forwards and backs. The excellent between-trial ICC values illustrate the high rank-order reliability of the included CMJ variables, which is an important consideration when compiling normative data. The positional differences in jump height, modified reactive strength index, and jump momentum, with accompanying moderate–large effect sizes, justified the separation of forwards’ and backs’ data before calculating the standardized T-scores. These findings echo those of previous studies [8,10]. The higher jump height values for the backs may be due to them having a lighter average body mass, whereas the higher jump momentum values for the forwards may be due to them having a heavier average body mass [8,10]. These positional differences in body mass, and therefore both jump height and momentum, emphasize the rationale for compiling position-specific normative data and benchmarks for these variables and echo the findings of previous work with respect to the importance of individualizing CMJ data reporting within rugby league to also account for the range of body masses within each position group [10]. 

The normative data presented in this study may be used by rugby league researchers and practitioners in several ways, provided that similar data acquisition methods have been employed. Firstly, the presented data may facilitate the interpretation of CMJ scores obtained by individual professional senior male rugby league players. For example, if a rugby league forward attains a CMJ height of 37 cm, then this may be interpreted as an above-average score (Figure 5). Secondly, the presented data may be used to set objective CMJ benchmarks for individual rugby league players. For example, a rugby league back who achieves a modified reactive strength index score of 0.42, which is below average (Figure 5), may be set a short-term modified reactive strength index target of 0.45 to equal the lower range of average. Once attained, a new benchmark may then be set for the rugby league back, such as a longer-term modified reactive strength index target of 0.53 to equal the lower range of above average (Figure 5). Thirdly, benchmarks may be set for academy-aged rugby league players who aspire to become professional senior players. Based on the CMJ results presented by McMahon et al. [16], academy (under 19s) rugby league players who demonstrated a mean jump momentum of 213 kg·m/s (presented as propulsion net impulse, which is equal to jump momentum, for the collective forwards and backs in McMahon et al. [16]) would be categorized as poor if compared with the forwards’ data, and below average if compared with the backs’ data, presented in this study (Figure 5). As jump momentum is highly related to sprint momentum [9] and sprint momentum is considered an important factor for rugby league players [35], rugby league practitioners working with academy players may wish to develop this quality as they transition to the senior level of competition. 

There are some aspects to this study that researchers and practitioners should be mindful of before using the presented normative data. The data were collected during the early preseason training period and so the presented values may not reflect the tested rugby league players’ best CMJ performance. The reason for conducting the testing at the beginning of the preseason phase is because this is the only equivalent period among different rugby league clubs. It is also the most common physical testing period adopted by strength and conditioning coaches [2]. Further into the season, each club adopts different training and competition schedules and has different volume-load accumulation, which would undermine the collation of data for a multi-club study such as this one. The jump height calculation used in this study was based on take-off velocity and not flight time. It is still common for the flight-time-derived estimate of jump height to be applied in practice and in published studies involving rugby league players, but this method usually overestimates jump height [26,36]. If using a force plate to test rugby league players, it is recommended that the take-off velocity method of calculating jump height should be applied where possible. Otherwise, exercise caution when comparing flight time estimates of jump height to the presented normative data in Figure 5. Finally, the presented CMJ data represent the mean score attained over the three recorded trials and not the single trial associated with the highest jump. This is because considering the mean of two to three CMJ trials, instead of the single best trial, has been recommended in previous studies to improve reliability [37,38,39]. 

While the overarching aim of this study was to produce rugby league position-specific normative data and create benchmarks for three primary CMJ outcome metrics, we would like to reiterate the notes in previous work by suggesting that CMJ force–time variables should be selected based on what is considered important to each individual player’s role during competitive matches [10]. For example, although forwards are generally heavier (i.e., based on the position mean) and are involved in more collisions than backs [40], particularly hit-up forwards [41,42], some individuals within the forward position may be lighter than some backs from the same squad and so more emphasis might be placed on CMJ height and the modified reactive strength index than jump momentum in such cases. The normative data and performance bands presented in Figure 5 are, therefore, only meant to facilitate rugby league practitioners’ decision making. We recommend that practitioners should interpret CMJ data on an individual player basis and, where possible, consider other force plate data from maximal strength (e.g., via the isometric mid-thigh pull) and plyometric (e.g., rebound jumps) tests too. Some examples of how rugby league practitioners may consider reporting key CMJ variables for specific players are presented in a recent study [10]. Additionally, aside from CMJ data being useful from a strength and conditioning perspective, they have also been shown to be highly correlated (*r* = 0.38–0.62) with better sprint and tackle performances during rugby league match play [13,14]. Furthermore, a recent principal component analysis of the CMJ revealed that it was a significant predictor (*R*^2^ = 0.19) of post-contact meters attained by rugby league players [43]. Warranted future research avenues, therefore, include combining multiple force plate test data for rugby league players to inform holistic training priorities to enhance both general athleticism and match-specific outcomes, and exploring how to effectively report primary variables from multiple tests in a visual and intuitive manner.

## Figures and Tables

**Figure 1 sensors-22-08669-f001:**
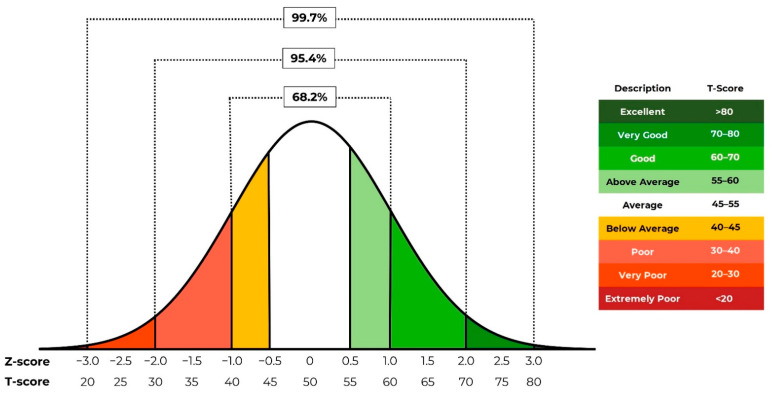
A typical normal data distribution curve with corresponding Z-scores and T-scores and integrated traffic light system example (T-score bands and qualitative descriptions are shown in the key to the right).

**Figure 2 sensors-22-08669-f002:**
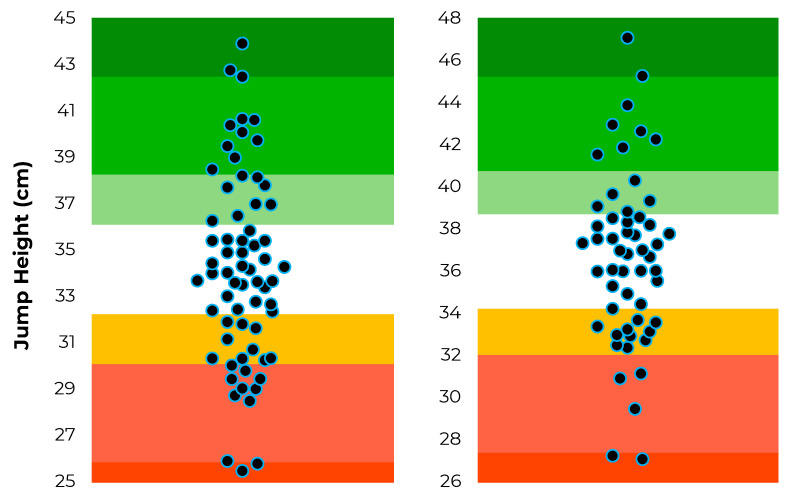
Distribution of jump height scores (represented by the individual dots) and corresponding traffic light system based on T-score thresholds for rugby league forwards (**left**) and backs (**right**).

**Figure 3 sensors-22-08669-f003:**
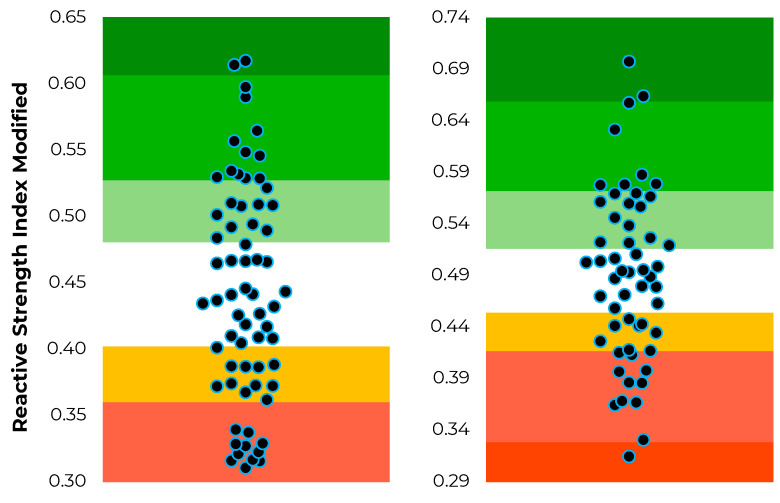
Distribution of modified reactive strength index scores (represented by the individual dots) and corresponding traffic light system based on T-score thresholds for rugby league forwards (**left**) and backs (**right**).

**Figure 4 sensors-22-08669-f004:**
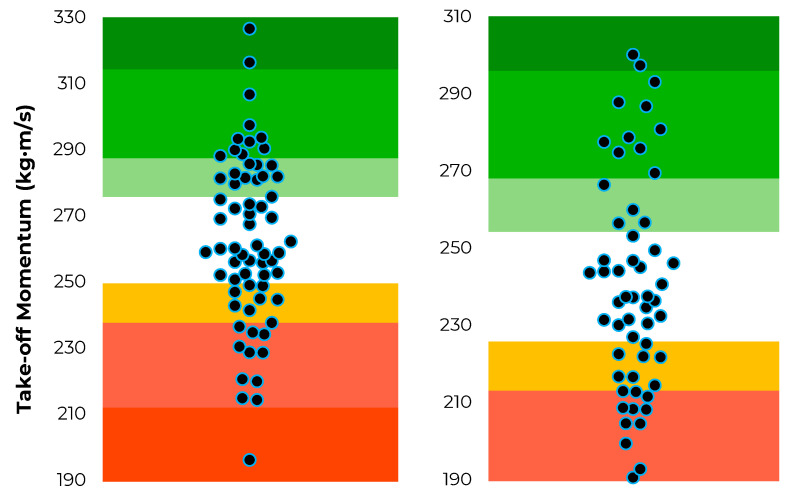
Distribution of jump momentum scores (represented by the individual dots) and corresponding traffic light system based on T-score thresholds for rugby league forwards (**left**) and backs (**right**).

**Figure 5 sensors-22-08669-f005:**
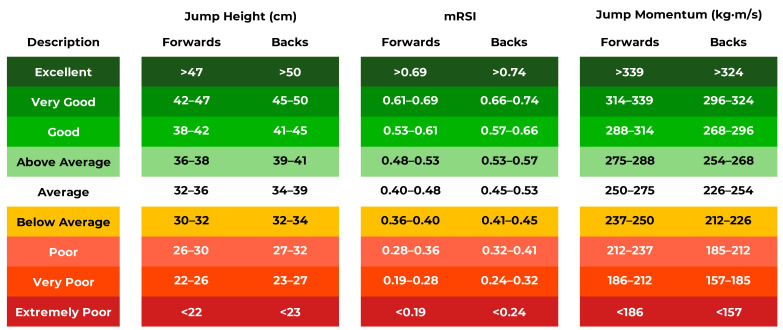
Countermovement jump normative data for rugby league forwards and backs with T-score bands and accompanying qualitative descriptions and traffic light system. mRSI = modified reactive strength index.

## Data Availability

Not applicable. All individual data points are shown in the figures referenced in the manuscript in addition to the normative data performance bands (Figure 5), which were directly calculated from the mean and standard deviation of each metric (see Section 2).

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
