# Peer review of "Force Plate-Derived Countermovement Jump Normative Data and Benchmarks for Professional Rugby League Players"

_sensors, 2022, doi:10.3390/s22228669_

Round 1
Reviewer 1 Report
First of all, I would like to thank the authors for the presented results of their investigation of high-level athletes. I especially appreciate such kind of effort bearing in mind how it is not easy to organize research on this population. Also, I would like to thank the editor for the opportunity to review this manuscript.
The manuscript entitled “Force plate-derived countermovement jump normative data and benchmarks for professional rugby league players” tends to establish countermovement jump normative data and objective criteria for professional rugby league forwards and backs. In my opinion, the authors present an interesting topic that falls within the aims and scope of the Sensors, special issue Applications, Wearables and Sensing Technology in Sports and Physical Activity. Since countermovement jump has been studied in various populations, as well as in the rugby players, new insights on this topic are always welcome.
The work has a good outline and is easy to read, although there is room for improvement in certain details. I would suggest that the authors consider improving the manuscript by highlighting the stage of preseason training periods through the Introduction section (not just in the limitation). I believe this is important since CMJ results vary during the season, and the results of this research actually suggest the condition that would be expected from athletes at the beginning of the preseason. Further, I would like the authors to consider the importance of the CMJ not just from the strength and conditioning coaches' perspective but also from a competition success standpoint. One or two sentences could enrich the manuscript.
Lines 228-234: I would suggest restructuring these two sentences. This way, it seems that body mass could be more important than the specific players' position requirements.
Lines 247-251: In line 249, there is an opening bracket, but the closing is missing. In line 251, there is a full stop after [16] that should not be placed here.
Author Response
Reviewer 1 Comments:
First of all, I would like to thank the authors for the presented results of their investigation of high-level athletes. I especially appreciate such kind of effort bearing in mind how it is not easy to organize research on this population. Also, I would like to thank the editor for the opportunity to review this manuscript.
The manuscript entitled “Force plate-derived countermovement jump normative data and benchmarks for professional rugby league players” tends to establish countermovement jump normative data and objective criteria for professional rugby league forwards and backs. In my opinion, the authors present an interesting topic that falls within the aims and scope of the Sensors, special issue Applications, Wearables and Sensing Technology in Sports and Physical Activity. Since countermovement jump has been studied in various populations, as well as in the rugby players, new insights on this topic are always welcome.
Authors’ response: many thanks for your time in completing your review of our work. We appreciate your constructive comments and have responded to them below.
The work has a good outline and is easy to read, although there is room for improvement in certain details. I would suggest that the authors consider improving the manuscript by highlighting the stage of preseason training periods through the Introduction section (not just in the limitation). I believe this is important since CMJ results vary during the season, and the results of this research actually suggest the condition that would be expected from athletes at the beginning of the preseason.
Authors’ response: thank you for your comment. We have added the below to the second paragraph of the introduction section:
“It is worthwhile noting that each of these studies involved data collection within the first two weeks of the pre-season period, which may not reflect the rugby league players’ best CMJ performances but provides a comparable stage of the year when conducting multi-club studies”.
Further, I would like the authors to consider the importance of the CMJ not just from the strength and conditioning coaches' perspective but also from a competition success standpoint. One or two sentences could enrich the manuscript.
Authors’ response: thank you for your comment. We have added the information below the discussion session (final paragraph): “
Also, aside from CMJ data being useful from a strength and conditioning perspective, it has also been shown to be highly correlated (r = 0.38-0.62) with better sprint and tackle performances during rugby league match play [13, 14]. Furthermore, a recent principal component analyses of the CMJ revealed that it was a significant predictor (R2 = 0.19) of post-contact meters attained by rugby league players [40]. Warranted future research avenues, therefore, include combining multiple force plate test data for rugby league players to inform holistic training priorities to enhance both general athleticism and match specific outcomes, and exploring how to effectively report primary variables from multiple tests in a visual and intuitive manner.
Lines 228-234: I would suggest restructuring these two sentences. This way, it seems that body mass could be more important than the specific players' position requirements.
Authors’ response: thank you for your comment. We have added some extra context to the end of the second sentence that you highlighted to illustrate that body mass can also vary within each positional group and that this should be considered via individualising CMJ reports for rugby league players (please see the below).
“These positional differences in body mass and therefore both jump height and momentum, emphasize the rationale for compiling position-specific normative data and benchmarks for these variables and echo previous work with respect to the importance of individualizing CMJ data reporting within rugby league to also account for the range of body masses within each position group [10].”
Lines 247-251: In line 249, there is an opening bracket, but the closing is missing. In line 251, there is a full stop after [16] that should not be placed here.
Authors’ response: thank you for spotting this. We have added the closing bracket and deleted the full stop after [16].
Reviewer 2 Report
Manuscript: Force plate-derived countermovement jump normative data and benchmarks for professional rugby league players.
General comments:
1. I think you have missed an opportunity here with the data. Why not calculate other variables? i.e. max positive and negative power. Relative impulse during particular phases (i.e. unweighted, stretch etc…review Sole et al., 2018). Max force during contact time. Max landing force. There are a lot more variables that could have been calculated and provided more relevant information to researchers and strength and conditioning coaches. Suggest completing the analysis for either this manuscript or a future one.
Sole, C.J., Mizuguchi, S., Sato, K., Moir, G.L., & Stone, M.H. (2018). Phase characteristics of the countermovement jump force-time curve: A comparison of athletes by jumping ability. Journal of Strength and Conditioning Research, 32(4), 1155-1165.
Abstract:
No comment
Introduction:
Page 1, line 33 - Can the authors explain what they mean by ‘which is used as a proxy for power’? I don’t believe you can use JH as a substitute for power. A reference needs to be used for such a statement. Suggest removing this statement as it holds benefit to the manuscript.
Page 1, line 34-37 - The sentence of what equipment is being used needs to be linked to the JH. Suggest writing. ‘To obtain the jump heigh metric, a recent survey conducted ……athletes. The benefit of the fore plate permits the….more interest jumping mechanics, such as power and impulse’.
Page 2, line 41 – Can the authors provide an example of what is the industry gold standard systems? Are these not force platforms?
Page 2, line 42 – You are repeating aspects of paragraph one. Need to review the intent of paragraph one. CMJ and force platforms in the general sense. Then start discussing Rugby League and CMJ in paragraph two.
Page 2, line 52 – ‘for professional rugby league backs’…you need to put in the comparison. i.e. ‘…backs compared to forwards’.
Methods:
Page 4, line 113 – You are using subjects and participants. i.e in page 3 line 110 you use ‘Subjects’ then in 113 you participants. Be consistent.
Results:
Page 6, line 191 – I suggest that the use of ‘significantly’ isn’t used. Instead ‘The forwards were taller than the back (P = 0.05) with ….’. The P value already covers if the data indicated a significant difference. Also, I think you need to put the data in here again. Going back to the methods to check the difference between the mean is a bit annoying. At least put in the difference.
Discussion:
Page 8 line 228-230 – ‘The higher jump height values for the backs may be due to them having a lighter body mass, whereas the higher jump momentum values for the forwards may be due to them having a heavier body mass.’ – Can the authors provide a reference for this statement? It could have been impulse during the unweighted phase to the net impulse that was the cause of the difference. When you have examined the jumping mechanics in a simplistic form there could be a lot of causes that are being missed.
Author Response
Reviewer 2 Comments:
Authors’ response: many thanks for your time in completing your review of our work. We appreciate your constructive comments and have responded to them below.
General comments:
I think you have missed an opportunity here with the data. Why not calculate other variables? i.e. max positive and negative power. Relative impulse during particular phases (i.e. unweighted, stretch etc…review Sole et al., 2018). Max force during contact time. Max landing force. There are a lot more variables that could have been calculated and provided more relevant information to researchers and strength and conditioning coaches. Suggest completing the analysis for either this manuscript or a future one.
Sole, C.J., Mizuguchi, S., Sato, K., Moir, G.L., & Stone, M.H. (2018). Phase characteristics of the countermovement jump force-time curve: A comparison of athletes by jumping ability. Journal of Strength and Conditioning Research, 32(4), 1155-1165.
Authors’ response: thank you for your comment.
In line 59, we wrote: “…only CMJ outcome variables […] differentiated between rugby league forwards and backs rather than […] selection of strategy variables [10]. In the study shown in the manuscript as citation 10, it was reported that none of the included variables calculated during the countermovement phase (i.e., unweighting and braking) discriminated between rugby league forwards and backs. Indeed, braking phase variables (such as peak force, peak power, net impulse etc.) did not discriminate between senior levels of play (i.e., players from the top two tiers of English rugby league) or academy vs. senior players, whereas the outcome variables included in the present study did (citations shown below).
- Int J Sports Physiol Perform. 2017 Jul;12(6):803-811. doi: 10.1123/ijspp.2016-0467.
- J Strength Cond Res. 2022 Jan 1;36(1):226-231. doi: 10.1519/JSC.0000000000003380.
This is why in lines 63-67, we wrote: “…from the collective results of these studies [8, 10], it may be suggested that CMJ scale of reference values for rugby league forwards and backs should be determined independently (i.e., position-specific) and based on primary outcome variables, such as jump height, reactive strength index modified and jump momentum.”
We have since added more context in the methods section (lines 155-158), as follows: “Mean and peak propulsion power were not included in this study, despite them being reported in a previous similar study [8]. This is because mean propulsion power is highly correlated (R2 = 0.81) with modified reactive strength index [30] and peak propulsion power is high correlated (R2 = 0.69) with jump height [31].”.
Ultimately, for benchmarking purposes, we must include variables that have been shown to discriminate between levels of play and where it is clear that a larger number is better (which is not straight forward when it comes to variables in the countermovement phase).
Introduction:
Page 1, line 33 - Can the authors explain what they mean by ‘which is used as a proxy for power’? I don’t believe you can use JH as a substitute for power. A reference needs to be used for such a statement. Suggest removing this statement as it holds benefit to the manuscript.
Authors’ response: thank you for your comment. We agree that jump height cannot (and, therefore, should not) be used as a proxy for power. We intended to highlight that this is, incorrectly, often done but for the sake of avoiding any confusion, we have deleted ‘which is used as a proxy for power’ from the sentence in line with your suggestion.
Page 1, line 34-37 - The sentence of what equipment is being used needs to be linked to the JH. Suggest writing. ‘To obtain the jump heigh metric, a recent survey conducted ……athletes. The benefit of the force plate permits the….more interest jumping mechanics, such as power and impulse’.
Authors’ response: thank you for your comment. We have amended the sentence to the following, in order to make the point clearer in line with your recommendation: “Results of a more recent survey conducted with Argentinean strength and conditioning coaches working in Rugby Union showed that 35% used electronic jump mats to assess jump height, whereas 17% used force plates [4], with only force plates permitting the calculation of impulse applied to the body’s center of mass from the initiation of the jump through to take-off.”
Page 2, line 41 – Can the authors provide an example of what is the industry gold standard systems? Are these not force platforms?
Authors’ response: thank you for your comment. Yes, we meant industry gold standard force plate systems (i.e., in-ground laboratory-grade force plates) and have clarified this in the revised sentence.
Page 2, line 42 – You are repeating aspects of paragraph one. Need to review the intent of paragraph one. CMJ and force platforms in the general sense. Then start discussing Rugby League and CMJ in paragraph two.
Authors’ response: thank you for your comment. This is still part of paragraph one, but we think we know where the confusion has come from. In the second part of paragraph one, we are referring to the rise in experimental studies involving CMJ testing of rugby league players. In the first part, we summarised the results of recent practitioner surveys. We think that “experimental” was the key word omitted from the sentence you commented on and, as such, we have since added it to line 43.
Page 2, line 52 – ‘for professional rugby league backs’…you need to put in the comparison. i.e. ‘…backs compared to forwards’.
Authors’ response: thank you. We have amended the sentence in line with your suggestion.
Methods:
Page 4, line 113 – You are using subjects and participants. i.e in page 3 line 110 you use ‘Subjects’ then in 113 you participants. Be consistent.
Authors’ response: thank you for spotting this. We have change to subjects to participants for consistency, as you suggested.
Results:
Page 6, line 191 – I suggest that the use of ‘significantly’ isn’t used. Instead ‘The forwards were taller than the back (P = 0.05) with ….’. The P value already covers if the data indicated a significant difference. Also, I think you need to put the data in here again. Going back to the methods to check the difference between the mean is a bit annoying. At least put in the difference.
Authors’ response: thank you. We have amended the whole paragraph by removing every inclusion of the word ‘significantly’ for consistency with your recommendation for removal of its use in the first sentence. We also included the descriptive data for body mass and height in the results section, as you suggested.
Discussion:
Page 8 line 228-230 – ‘The higher jump height values for the backs may be due to them having a lighter body mass, whereas the higher jump momentum values for the forwards may be due to them having a heavier body mass.’ – Can the authors provide a reference for this statement? It could have been impulse during the unweighted phase to the net impulse that was the cause of the difference. When you have examined the jumping mechanics in a simplistic form there could be a lot of causes that are being missed.
Authors’ response: thank you for your comment. Yes, we have added two citations to support the statement (please see the updated sentence below) and we have rationalised further above the reasons why we have kept the analysis for this study simplistic.
“The higher jump height values for the backs may be due to them having a lighter body mass, whereas the higher jump momentum values for the forwards may be due to them having a heavier body mass [8, 10].”